# Deep Learning Applications for Dyslexia Prediction

**Norah Dhafer Alqahtani** [1,2,*], **Bander Alzahrani** [1] and **Muhammad Sher Ramzan** [1,*]

1. Faculty of Computing and Information Technology, King Abdulaziz University, Jeddah 21589, Saudi Arabia
2. Information Systems, King Khaled University, Abha 61421, Saudi Arabia
* Correspondence: nhalqahtani@stu.kau.edu.sa (N.D.A.); msramadan@kau.edu.sa (M.S.R.)

**Abstract:** Dyslexia is a neurological problem that leads to obstacles and difficulties in the learning process, especially in reading. Generally, people with dyslexia suffer from weak reading, writing, spelling, and fluency abilities. However, these difficulties are not related to their intelligence. An early diagnosis of this disorder will help dyslexic children improve their abilities using appropriate tools and specialized software. Machine learning and deep learning methods have been implemented to recognize dyslexia with various datasets related to dyslexia acquired from medical and educational organizations. This review paper analyzed the prediction performance of deep learning models for dyslexia and summarizes the challenges researchers face when they use deep learning models for classification and diagnosis. Using the PRISMA protocol, 19 articles were reviewed and analyzed, with a focus on data acquisition, preprocessing, feature extraction, and the prediction model performance. The purpose of this review was to aid researchers in building a predictive model for dyslexia based on available dyslexia-related datasets. The paper demonstrated some challenges that researchers encounter in this field and must overcome.

**Keywords:** dyslexia detection; dyslexia classification; feature extraction; diagnosing dyslexia; machine learning; deep learning

## 1. Introduction

Dyslexia is a common learning difficulty that people encounter throughout their learning journey, which affects the reading, writing, spelling, fluency, word decoding, and dictation processes. However, it is not related to an individual's level of intelligence. The term originates from the ancient Greek, with the prefix "dis" referring to disorder and the root "lexia" to language. Hence, dyslexia signifies a language defect or disorder [1]. Many children with this disorder have normal intelligence and receive appropriate education and parental support but have difficulty with learning certain skills. Today, dyslexia is the most frequent childhood learning disorder, accounting for up to 80% of all identified learning disabilities [2].

The World Federation of Neurology identifies dyslexia as a disturbance where the child's spelling, writing, or reading skills fail to meet predicted levels based on age and intellectual performance despite attending school regularly [3]. From a neuropsychological approach, these disorders result from one or more malfunctioning learning-related brain systems [3], where the functions of the left hemisphere are imbalanced, such as impairment in the area concerned with short-term memory, motor skills, visual perceptions, language processing, auditory, speed, and speaking (Figure 1).

The early diagnosis of suspected dyslexia in children is essential as it will increase the likelihood of the dyslexic child benefiting from effective intervention programs and improving his or her abilities [4]. Traditional tests for diagnosing dyslexia depend on the evaluation of reading words and text, writing, and working memory. Experts in this field have defined and normalized the scores of these tests. Some screening tests include CASL, TAPS, CTOPP-2, WRMT, GSRT, and TEWL, which have become available on the Web due to ICT that authorized the digitalization of these tests [5]. ICT refers to a varied set of

resources and technological tools utilized to create, transfer, store, interchange, or share information. All these tests are available for subscription under specific criteria.

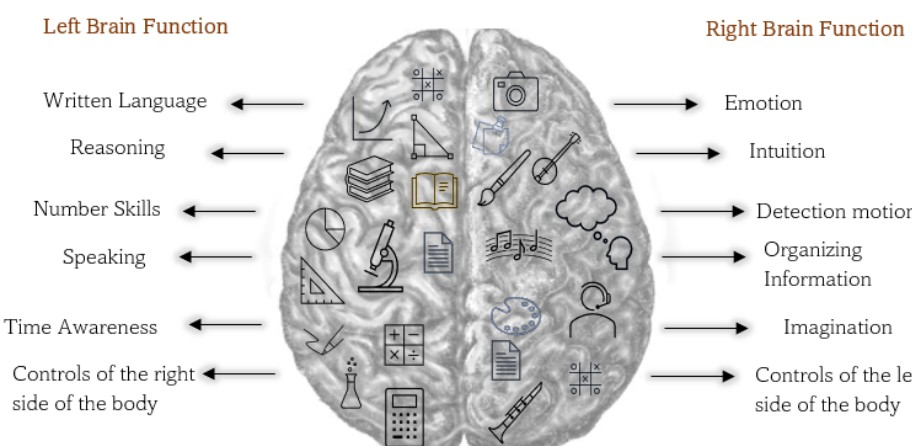

**Figure 1.** This image shows that any disorder in the brain's left hemisphere would lead to important difficulties in a person's ability to read and write and some other skills.

Neurological aspects have also been considered in modern technologies for diagnosing dyslexia, helping to increase detection accuracy and reliability. Medical devices enable the observation of the brain structure of dyslexics [6]. Habib [7] proposed three triggering mechanisms of dyslexia: attentiveness disturbance (visual–attentional dyslexia), language disturbance (phonological dyslexia), and motor disturbance (a dyspraxic form of dyslexia).

In "phonological dyslexia", during the reading, fMRI displays paralysis of three regions that are concerned with language production and grasping (Geschwind's, Wernicke's, and Broca's territory) [8]. Additionally, compared to controls, DTI reveals white matter variation in the language area [9]. In visual attentional dyslexia, when dyslexic participants are tasked to recognize congruous stimuli pairs, fMRI displays a separation between the temporal visual system and parietal attentional system as well as a disconnection, in the left hemisphere, of the temporal and occipital zones [9]. In the "dyspraxic form of dyslexia", there is inactivation in the cerebellum–ventral frontotemporal and cerebellum–dorsal frontoparietal pathway [10]. Besides fMRI, the surface measurement of brain potential, known as EEG, assists in identifying brain activation patterns. During spelling tests, phoneme deletion, the rapid naming of letters, and articulation, increased vigor is observed in the theta and delta EEG frequency bands in the frontal and right temporal zones in dyslexics [11].

Researchers have suggested different ML methods for predicting dyslexia in children utilizing datasets related to dyslexics. Such datasets can be acquired from medical and educational institutions as well as through special games constructed specially for them. For example, eye tracking, brain imaging, EEG, test scores, and handwriting have been used in dyslexia prediction.

AI refers to the development of machines and systems to enable them to implement functions and tasks that demand human intelligence, such as translation, decision making, visual perception, and speech recognition. ML is considered a part of AI, which concentrates on the evolution of computer programs that use different datasets to learn for themselves. DL is a subset of ML that seeks to simulate the human brain, allowing DL systems to cluster data and make incredibly accurate predictions. The literature has demonstrated the success of ML methods in classification problems, particularly the classification of diseases. Moreover, ML methods have been found to have outstanding accuracy for diagnosing dyslexia [12]. However, traditional ML methods fail to use raw data to implement these tasks [13]. To overcome this obstacle, ML models have used a layered learning approach, known as the DL. The general difference between DL models and traditional ML models is that DL models do not require the engineering step of feature extraction, which is inherent to conventional models, as they automatically learn abstract hierarchical feature

representations from data [14]. ML might be utilized to accomplish specific tasks, for instance, the identification of objects in images, speech-to-text transformation, and isolation of items in categories [13].

DL utilizes ANNs that simulate the brain's function. Since ANNs were introduced, they have undergone considerable alteration and development; however, the basic principle has remained the same [13]. The ANN structure essentially comprises nodes and edges that, together, create a structure that resembles a neuron and are in charge of transmitting the information [13]. It consists of three strata: input strata, several hidden strata, and output strata. In each layer, every node is linked to every other node in the next layer. The network becomes deeper as the number of hidden layers increases, leading to a DNN model. A DNN converts raw inputs to helpful features. Subsequent layers elicit a group of features that are considered more abstract and help to achieve a desired task. The essential processes in DL are feature extraction and selection [13]. DNNs have various architectures. A basic one is a feed-forward network, in which the structure has links among layers in a single direction (forward), and there are no existing loops or cycles in the whole structure. Other architectures include particular functions, such as RNNs, which are implemented with sequential datasets, whereas CNNs are implemented with grid-structured data-like images. MLP and CNN models have been applied at different times and places to reveal and classify dyslexia utilizing special data related to dyslexic children. This review paper focuses on the use of advanced ML methods (DL) to help in diagnosing dyslexia disorder and constructing a predictive model for prediction. The following points are investigated:

- The types of datasets that are used by prediction models.
- Different DL models that are utilized to predict dyslexia disorder.
- The performance of DL models in dyslexia prediction.

To the best of our knowledge, this is the first review to investigate current works that specifically utilized DL models to diagnose dyslexia. This will allow us to investigate the points above, seeking to find a new way to predict dyslexia. Previous reviews on dyslexia prediction have focused on ML models in general, not on DL models.

Following this initial introduction, Section 2 provides an overview of related works, both review papers and systematic reviews, and illustrates the difference between previous reviews and the current work. Section 3 discusses the search strategy and explains the stages of the article selection process, including the criteria. Section 4 provides a detailed explanation of every article in the results section, including information on dataset acquisition, dataset preprocessing, feature selection, and model prediction performance. Section 5 lists and discusses some challenges related to scanning the articles, and Section 6 presents concludes the paper.

## 2. Related Work

Several surveys and review papers have addressed the use of ML in dyslexia disorder classification. During the research process, we found nine papers that were related to the diagnosis of dyslexia disorders using ML methods. In a review paper [15], the research covered several dimensions related to dyslexia prediction with ML methods and image processing techniques. In addition, regarding design assessment tests and tools for prop dyslexics, most of the research utilized ML techniques to predict dyslexia. A survey paper [16] summarized the techniques for diagnosing dyslexia that use ML approaches. It screened 13 studies that applied ML methods, and only one of these studies applied DL methods for dyslexia classification. In three review papers [6,17,18], the authors conducted literature surveys focused on ML methods utilized in dyslexia prediction. The surveys did not include studies on dyslexia prediction based on handwriting datasets, which started in 2019. A systematic review [19] has focused on ML and DL methods that have been utilized for identifying dyslexia and its biomarkers and concluded that Support Vector Machine (SVM) is the most frequently applied ML method for identifying and predicting dyslexia. Moreover, it noted that the utilization of DL algorithms is still in its infant stage. An extensive review [20] focused not only on the prediction of dyslexia disorder using

ML methods but also included studies on the prediction of attention deficit hyperactivity disorder. Ahire et al. [21] conducted a comprehensive review on the classification of dyslexia using ML methods, focusing specifically on studies that used EEG signals for classification. They found that SVM outperformed other ML methods in EEG signal classification. Poornappriya and Gopinath [22] considered studies that implemented ML methods for dyslexia prediction in addition to studies that helped dyslexics to improve their reading and other skills. Their review was different from the other review papers, as it focused on the utilization of DL methods in the prediction of dyslexia using different dyslexia-related datasets.

### 3. Research Strategy

In this paper, a critical review of the literature has been performed to collect a broad variety of studies that used DL methods for dyslexia prediction and classification and to investigate the research points listed above. The identification and selection of pertinent articles followed the Preferred Reporting Items for Systematic Reviews and Meta-Analyses (PRISMA) guidelines [23]. As shown in Figure 2, the article selection process included three phases: identification, screening, and inclusion phases.

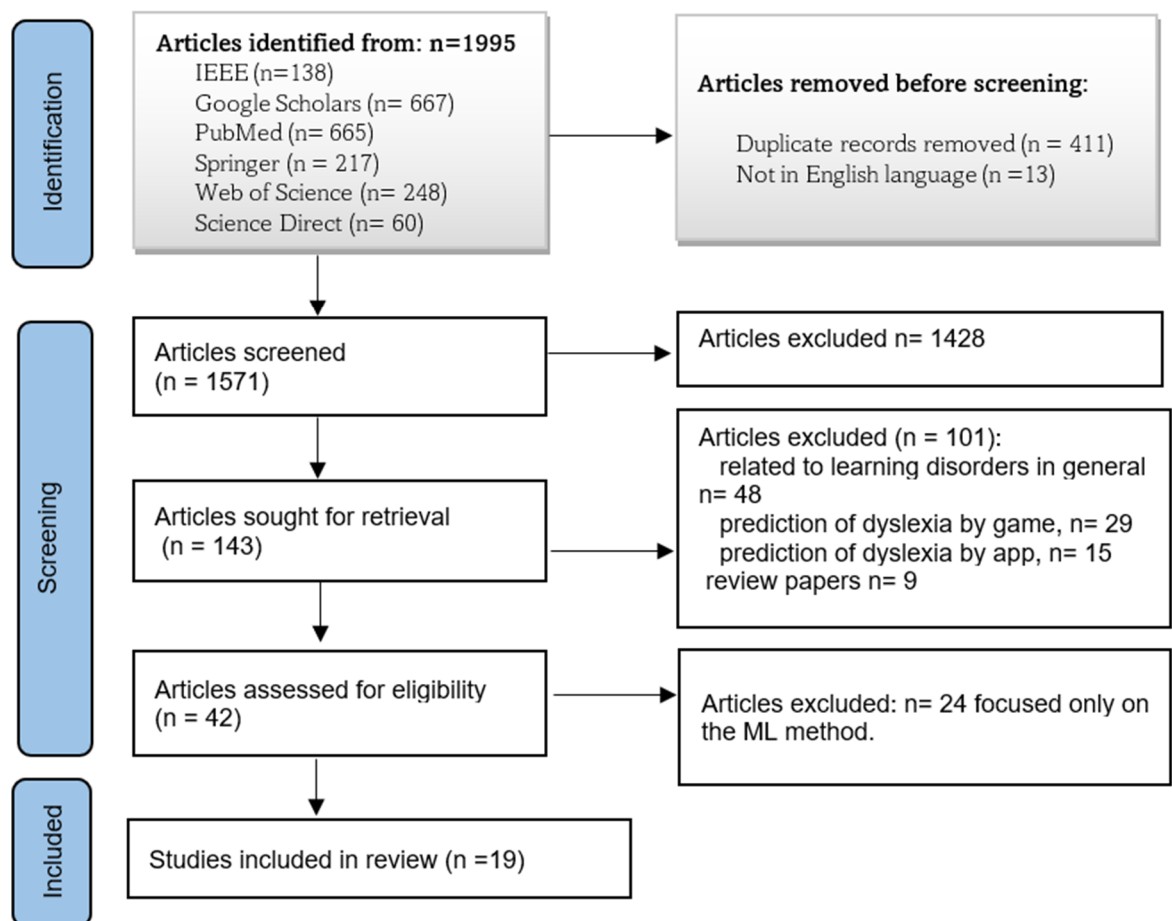

**Figure 2.** PRISMA flow diagram for selecting 19 definitive articles.

In the identification phase, we searched for appropriate articles from 2010 to 2022 in different databases (Figure 2), including IEEE Xplore, Google Scholar, PubMed, Springer, Web of Science (WoS), and Science Direct. The following keywords were used in the search queries: "Dyslexia classification", "Dyslexia Prediction", "ML in Dyslexia", "Deep Learning in Dyslexia", "Diagnosis Dyslexia", and "CNN in Dyslexia". The search produced 1995 articles, of which 138 were from IEEE, 667 from Google Scholar, 665 from PubMed,

217 from Springer, and 248 and 60 from WoS and Science Direct, respectively. We excluded 13 non-English articles and 411 duplicate articles.

In the screening phase, there were 1571 articles we needed to screen. The reviewers excluded 1428 irrelevant articles. Then, of the remaining 143 articles, 101 articles were excluded for different reasons. Some of the articles focused on dyslexia with other learning disorders, some focused on predicting dyslexia using games and applications, and some others were review papers. Then, the screening advanced after fulfillment of the inclusion criteria.

As shown in Table 1, the inclusion criteria were as follows: (1) articles released between 2010 and 2022 in English, (2) articles that utilized a DL method or combinations with traditional ML for the identification of dyslexia, and (3) articles that utilized datasets related to dyslexia.

**Table 1.** Inclusion and exclusion criteria.

| Inclusion Criteria | Exclusion Criteria |
| --- | --- |
| Articles released between 2010 and 2022 in English (AND) | Articles not relevant to dyslexia classification (AND) |
| Articles that utilized DL methods (OR) DL methods combined with traditional ML for the identification of dyslexia | Articles that used only traditional ML methods to prediction dyslexia (AND) |
| (AND) Articles that utilized datasets related to dyslexia | Articles that did not meet the inclusion criteria |

Subsequently, in the inclusion phase, the articles were reviewed for further consideration of their eligibility, based on the criteria shown in Table 1. Ultimately, only 19 articles were chosen for critical review. Following a previous study [13], we included ANN and MLP models in this review, where ANN include DL feed-forward networks as well as MLP. Figure 3 illustrates the number of articles selected per year from 2010 to 2022.

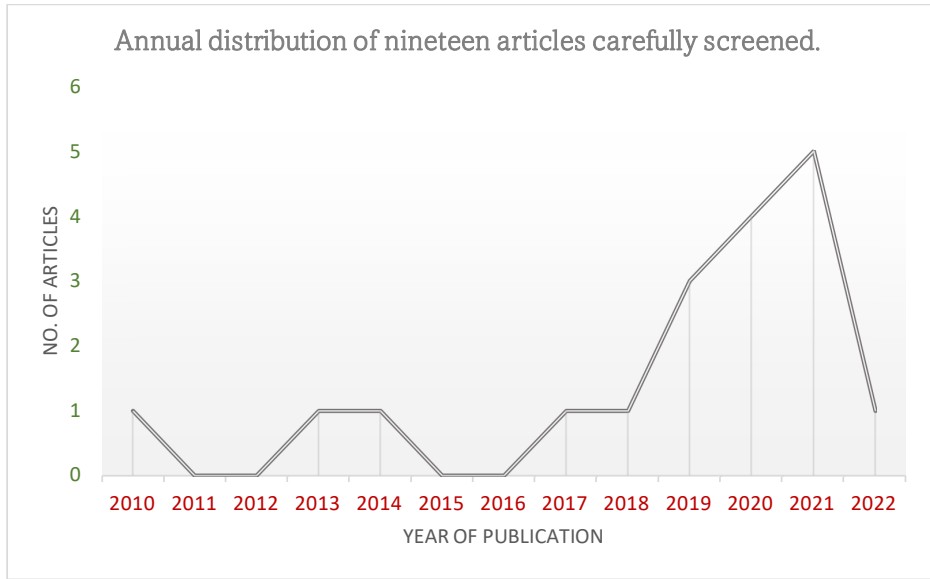

**Figure 3.** Annual distribution of the 19 articles that were carefully screened.

## 4. Research Results

Dyslexia prediction using DL models involves several steps, starting from acquiring the datasets to evaluating the prediction models. This can be seen in the articles selected for review.

### 4.1. Data Acquisition

The first step in predicting dyslexia disorders using DL methods is the acquisition of datasets related to dyslexia. As mentioned above, brain imaging and eye-tracking data in addition to traditional data (e.g., test scores and time consumption when performing specific tasks) can be used to diagnose dyslexia. In [19], the dyslexia datasets were divided into three classes. The first category was related to behavioral symptoms in dyslexics, the second category was related to brain imaging, and the last was related to eye movements during cognitive tasks. The behavioral symptoms of dyslexics manifest through reading, writing, working memory, phonological awareness, and facial reactions throughout the learning process. Reading and writing are inextricably linked, and children who struggle to read also struggle to write. Hence, dyslexics have problems when writing letters. For example, instead of starting from the top, they start from the bottom when writing letters, inverting the letters, skewing some, and erasing frequently [24]. Spoon et al. [25] were the first to use handwriting datasets to predict dyslexia. They compiled a handwritten dataset on 88 subjects (11 from dyslexics and 77 from normal subjects), demonstrating the possibility of using these types of data for predicting dyslexia.

In another study, 100 handwriting samples were collected (22 dyslexia and 78 non-dyslexia) from children in grades K-6 [26]. Isa et al. [27] collected 30 handwriting samples from the ADM. The dataset included eight selected numbers and small letters (6, 7, 2, 5, c, b, p, and f). The main reason for selecting these numbers and letters is that they are the most commonly associated with writing errors among dyslexics. In India, Yogarajah and Bhushan collected 54 handwriting samples from the notebooks of children (18 females, 36 males) from first to fifth grade [28]. Fourteen Hindi words were selected from the children's notebooks, representing varying degrees of difficulty. Five words consisted of two letters, six words had vowel signs, and three were conjoined consonants words. Isa et al. [4,29] used a dataset collected from three sources: uppercase letters were taken from NIST Special Database 19, lowercase letters were taken from the Kaggle dataset, and testing data were collected on Seberang Jaya Primary School's dyslexic students. These two studies used data augmentation techniques to generate the dyslexic datasets, using noise injection and rotation techniques to generate reversal handwriting datasets. Kohli and Prasad [30] collected a dataset from the test scores of dyslexic students on reading, spelling, speech deficits, deficits in writing, and mathematical abilities as well as motor skills.

The second category of dyslexia datasets is related to brain imaging, where the modalities of brain imaging identify special behaviors and activations of the brains of participants in the studies [19]. EEG is a brain imaging modality that can be utilized to observe the brain's function through electrodes placed on the scalp. In the studies, the researchers examined "neurological" aspects to identify unique patterns in dyslexia [31]. For example, Karim et al. [32] collected EEG signal data from six participants between four and seven years of age (three dyslexic and three non-dyslexic). The normal children were selected randomly from various schools, while the dyslexic children were from the Titiwangsa and Ampang Hilir branches of the Dyslexia Association of Malaysia. In the KSA, Al-Barhamtoshy and Motaweh [33] compiled a dataset on 80 children between 7 and 13 years of age with the help of the "Brain training and consultation center". Normal children were randomly selected from the cities of Jeddah and Makah. In another study [34], a dataset was compiled on 32 children who were native Hebrew speakers in grades 6–7 (17 dyslexics and 15 skilled readers). In Spain, Ortiz et al. [35] collected a dataset on 48 native Spanish-speaking participants (32 skilled readers and 16 dyslexics). All of them were right-handed, did not suffer from hearing impairments, and had normal vision. Usman and Muniyandi [12] used MRI data, which is a diagnostic medical tool utilized to analyze alterations in the brain anatomy. They collected neuroimaging data on 45 individuals (19 dyslexics and 26 non-dyslexics) aged 15–23 years from Kaggle datasets. fMRI is a type of MRI that produces images of brain soft tissue with the highest resolution, which has been utilized to identify and analyze various regions in brain [36]. In another study [37], an MRI dataset of BOLD functional

images was collected through different reading tasks: a lexical decision task, semantic categorization task, and lexical orthographic matching. The datasets were collected from 55 Spanish children between 9 and 12 years of age, who were recruited from schools and the University Hospital of Cruces in Spain. Chimeno et al. [38] gathered 3D images from both DTI and fMRI scans of 52 schoolchildren between nine and 12 years of age at various times. In 2021, an fMRI dataset was collected on 32 Portuguese children, divided into 16 typical readers and 16 dyslexics aged 8–12 years [39]. All children were right-handed and matched for IQ, age, and sex.

The final dataset category relates to patterns of eye movement when performing cognitive tasks, as recent studies have used eye movements to differentiate between dyslexic and non-dyslexic individuals. Statistical measures have been used to identify features of children's eye movement through cognitive exercise. The authors in [40,41] used the same dataset containing raw eye movement data on 185 subjects, 88 at low risk of dyslexia and 97 with a high risk of dyslexia. These data were collected in 2016 [42] from a wide population of 2165 school children in second grade. Recently, Vajs et al. [43] collected and analyzed a dataset on 30 subjects aged 7–13 years, 15 of whom had dyslexia and 15 of whom were normal participants (11 male and 19 female), as they read a Serbian written text with 13 different color configurations.

One study used an EOG in the prediction of dyslexia [44]. EOG is a method that relies on the screening of the electrical potential for eye movements, which is helpful for analyzing various types of eye movements, such as saccades, smooth pursuits, vergence, blinks and gaze fixation. This study collected data on 33 children aged between 8 and 11 years of age (20 with dyslexia and 13 healthy; 17 female and 16 male). None of the dyslexic children had hearing or vision defects. Figure 4 illustrates the usage percentage of different datasets in the articles that employed DL methods to predict dyslexia.

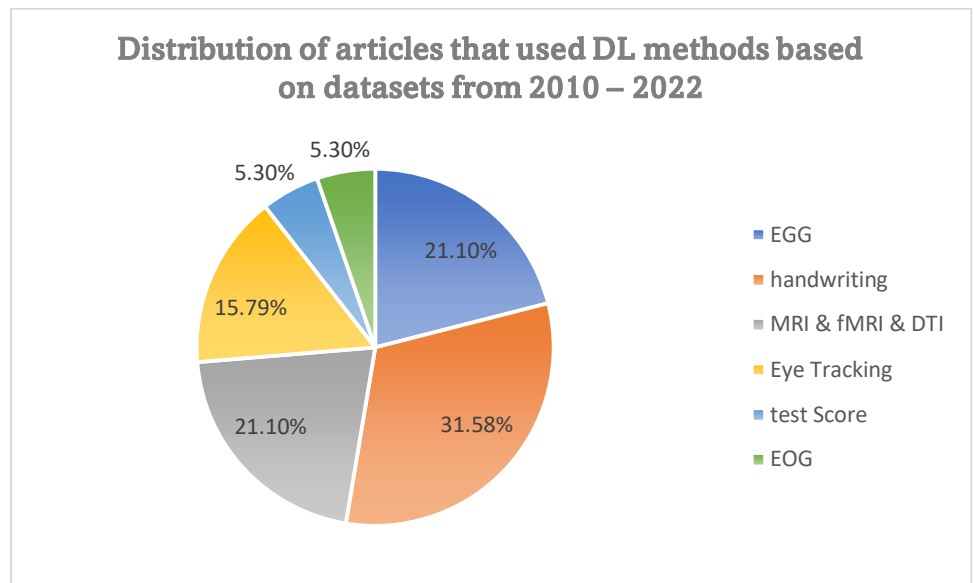

**Figure 4.** Distribution of articles that used DL methods based on datasets from 2010 to 2022.

### 4.2. Data Preprocessing

Datasets related to dyslexia need to be processed and prepared before used in DL techniques. The goal of this task is to enable the classifier to elicit the most related interpretable features from the preprocessed dataset. There are several objectives of preprocessing processes, including modulation, tissue segmentation, data normalization, smoothing, alignment with a particular image template, and data normalization [19]. The study by Kohli and Prasad [30] was different from the other studies in this review, as it utilized manual data preprocessing methods. One study [32] applied decimation and STFT to the fMRI signals, which were filtered using a bandpass procedure to remove noises and irrelevant artifacts.

In addition, an FSL instrument, the FMRIB software library, was employed for processing DTI and fMRI [38]. Usman and Muniyadi changed all of the T1w neuro-images (images of the brain) into a format that could be used by FreeSurfer software [12]. Then, the intensity of the images was adjusted so that they all had the same brightness and contrast. The normalization of intensity was based on histograms. A FSL FNIRT software instrument was utilized for non-rigid registration in the brain template of MNI152, and a Gaussian filter was used to minimize noise. In [37], the preprocessing step included a DICOM to NIFTI conversion, taking the "Digital Imaging and Communications in Medicine" file and converting it into a "Neuroimaging Informatics Technology Initiative" file. This conversion was necessary in order to use the data in SPM (Statistical Parametric Mapping) software to analyze the brain image data. Normalization and smoothing were applied in addition to the adjustment of head motion. In another study [39], the preprocessing of the task-based fMRI involved the correction of slice time and motion, smoothing, and normalization to a template of a voxel. Barhamtoshy and Motaweh [33] used standard algorithms to filter irrelevant records and noise from the dataset. In addition, they removed noise from EEG scans using a Fourier transform algorithm. Other authors used various wavelet transformation approaches to convert EEG scans into low-pass and high-pass filters [34]. Independent component analysis was used in the preprocessing of EEG signals to remove artifacts due to recording eye blinking signals along with EEG signals [35]. The studies that used the eye-tracking dataset to predict dyslexia focused on eliminating blinks and processing missing data [40,41,43]. The study that used EOG signals utilized the Butterworth bandpass filter to eliminate noise, followed by EOG segmentation. The aim of segmentation was to balance the dataset (dyslexic and healthy groups). With regard to a handwriting dataset, Spoon et al. [25,26] applied the DeepWriter concept [45] to create 50 random patches of handwriting features from each image in the handwriting dataset. Yogarajah and Bhushan [28] utilized the same concept to process Hindi letter datasets. Some studies included different steps for processing unstructured data (handwriting images), as shown in [27], including RGB to grayscale conversion, maximally stable extremal region, canny edges detector, stroke width filter, and morphology as a final step. Isa et al. [4,29] employed the same dataset. For preprocessing, they interchanged the foreground and background to minimize computational overhead when an image had a lower black point (value 0) than white point (value 1), which required more memory and power consumption in training the image. In addition, they cropped handwriting images to the actual writing part, thus resizing the images.

### 4.3. Feature Extraction and Selection

Before moving on to the classification stage, we need to be able to pull useful features from the acquisition dataset. Feature extraction refers to the process of converting raw data into numerical (or categorical) features that could be handled in classifier model without lost valuable data in original dataset [46]. The purpose of this operation to remove excrescent data and produce the most pertinent and highly informative features from the original ones. It is possible to extract features manually or automatically: Manual extraction demands defining and explaining the important features relevant for a given situation, as well as designing a method for extracting those features. In many circumstances, having a solid grasp of the context or domain can aid in making well-informed decisions regarding which features may be valuable.

Automated feature extraction utilizes skilled algorithms or deep networks to elicit features automatically from the dataset, with no human intervention required. An automatic feature extraction could be very helpful in assisting to move speedily from the original dataset to improve ML algorithms.

With the rise of DL, the first layers of deep networks have mostly taken the place of feature extraction. The features extracted for predicting dyslexia differed in the dyslexia-related datasets, as shown in Table 2.

**Table 2.** Feature extraction methods for dyslexia disorder detection utilizing Deep Learning Methods.

| Reference | Dataset | Feature Extraction Method and Selected Features |
|---|---|---|
| [30] | Test score | The study extracted features manually based on cognitive and evaluation test results. These included an IQ test, rapid naming test, evaluation of short-term memory, sequencing skills, and non-word reading to evaluate phonological coding skills. |
| [32] | EGG | Using a kernel density estimation process, brain activity features were extracted from EGG signals (353 features). The ksdensity () function in MATLAB was used to derive features depending on the normal kernel function. |
| [35] | | To extract features from the EGG signals, the study used SSA, which divides the raw signal into additive components representing various oscillatory manners. Five components were generated that might be utilized to explain the data. For each component, Pearson's correlation among various channels of the PSD of each singular SSA component was calculated. Pearson's correlation represents the degree of similarity between two channels, thus helping to differentiate between dyslexia and normal functioning. |
| [33] | | Brain electrical signal features were extracted from EGG signals utilizing Fourier transform algorithms and statistical functions. The algorithms used the rule-based model to filter non-related features and eliminate noise from the electrical signal records. |
| [34] | | The study used discrete wavelet transform techniques to extract the most beneficial ERP signals from EGG, which have a waveform linked both in frequency and time domain. The signal was divided into low pass and high pass. A group of temporal features were extracted from the low-pass portion such as latency, absolute amplitude, positive area, and entropy, while a group of statistical and spectral features were extracted from the high-pass portion such as mean, skewness ratios, standard deviation, and zero crossing rate. Moreover, some features relevant to the frequencies' structure were extracted such as spectral flatness measure, spectral centroid as well as power spectral density. |
| [41] | Eye Tracking | The various eye movement events for preprocessed data have been analyzed, such as saccades and fixations. Different features relevant to these events have been extracted using major statistical measures, dispersion, and approaches velocity-based. Two algorithms for feature selection have been used in this study, which are Recursive feature Elimination with Cross-Validation (RFE-CV) and PCA. |
| [40] | | In this study, the CNN model extracted, implicitly, substantial features scattered either in time or frequency from preprocessed eye-tracking data and nonlinearly bound them using machine learning to minimize detection error. |
| [43] | | The velocity features which extracted from eye-tracking events discriminate between dyslexic and control. |

**Table 2.** *Cont.*

| Reference | Dataset | Feature Extraction Method and Selected Features |
|---|---|---|
| [44] | EOG | The study used a 1D CNN model that generated feature maps through operations in layers. These feature maps contained significant features from the vertical and horizontal EOG signals, allowing the differentiation of dyslexic and normal readers. |
| [12] | MRI and fMRI | Features of the phonological and cognitive brain systems related to gray matter, white matter, and cerebrospinal fluid were extracted from fMRI utilizing CAT12 implemented in MATLAB. |
| [38] | | The study used FDT, BET, and TBSS to extract FA from DTS signals, and used two tools (BET and FEAT) to elicit features from fMRI. These two features (activation pattern and FA) are associated with speech, language, and lexical decisions. |
| [37] | | After preprocessing fMRI dataset, SMP12 has been implemented in MATLAP 2018b to extract cognitive features pertaining to grey and white matter and the volumetric biomarkers of cerebrospinal tissues. |
| [39] | | This study was considered unique due to its visualization of features, which differentiated dyslexic readers from normal ones, such as activation patterns of anterior right-hemisphere prefrontal areas as well as activation patterns in the left occipital and inferior parietal areas that distinguished groups based on brain networks related to lexical and phonological processes in reading. The feature extraction has been carried out in the layers of the LeNet-5 model. |
| [25,26] | Handwriting | These studies used behavioral and cognitive biomarkers (picture patches of handwriting features) to differentiate between dyslexic and normal readers. A CNN model extracted high features from processed handwriting images when implemented in Python. |
| [28] | | Hindi words have some prodigious features, such as diacritics (matras), conjoined consonants, and killer strokes (halants). The study used patches of handwriting features to differentiate between dyslexics and normal readers, which have been extracted in the CNN model |
| [27] | | The study used an OCR technology to identify letters in handwriting images. The rectified characters are displayed in the command box after picture extraction and character recognition, manually the total of correct detection was been calculated. |
| [4,29] | | These studies used a CNN model to extract features and predict dyslexia from handwriting images. The preprocessed handwriting images were fed to the CNN model, and a feature map was created through the convolution layer, which contained highly informative features. |

### 4.4. Classification and Performance of Deep Learning Models

As illustrated in traditional studies, ML methods have been used in the prediction of dyslexia, and recent advances in ML methods (DL) have resulted in outstanding accuracy. The use of DL models in dyslexia prediction varies based on the dataset used. For example, CNN models are appropriate for handling unstructured data, as shown in all studies

presented in this paper except [30], which dealt with numerical data. Although some studies used ANNs to deal with EGG datasets and achieved high accuracy (96%) [33], others achieved lower accuracy (78%) [34]. In addition, some studies that adopted handwriting datasets used CNN models and reported accuracies between 86% and 95% [4,28,29], while others achieved low accuracy (maximum 77%) [22,23,25]. Typically, the datasets were partitioned to train and test the prediction models. Most of the data (approximately more than or equal to 70% of all the data) were utilized for training, and the smaller part was allocated for testing model. Some studies allocated part of the dataset (might be 10% or 20%) as a validation dataset, which is a sample of data that is restricted from the process of the training model. These data are used to provide an estimate of the model's skill while the hyperparameters are being tuned. This division varied between studies. In refs. [12,29], 70% of the dataset was allocated for training, and the remainder was allocated for validation and testing. Some studies used a higher percentage. For example, refs. [39,41,43] allocated 80% for training, while study [40] allocated 90%. Recent studies have utilized ANN, MLP, and CNN models for dyslexia prediction and most have reported high accuracy in most studies. Figure 5 organizes the reviewed studies according to the DL algorithms adopted.

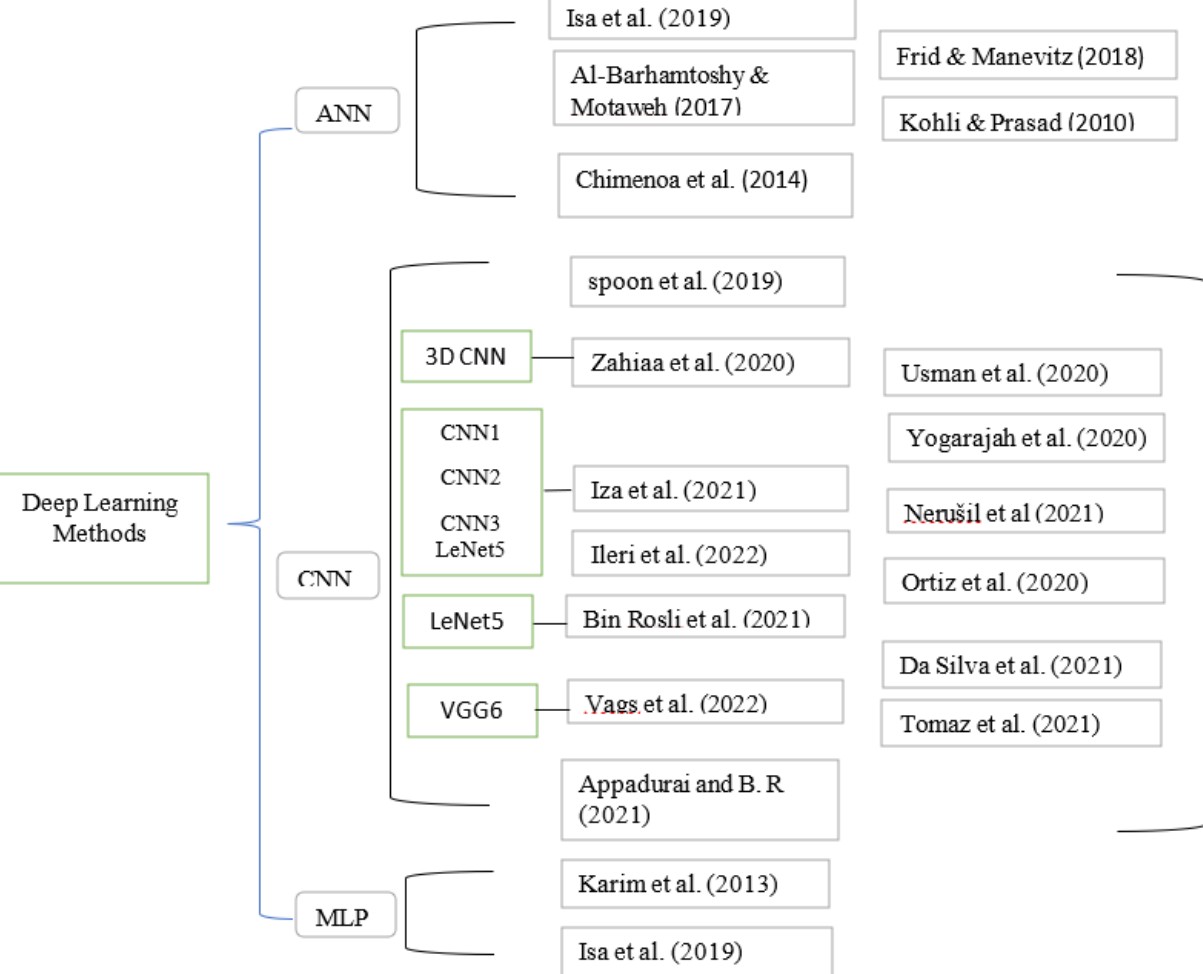

**Figure 5.** The reviewed studies organized according to the DL algorithms that each model adopted.

The ANN model suggested by the authors in [30] achieved an accuracy of 75% after ten-fold cross-validation when used with a dataset on pupils' performance acquired from a structured questionnaire. In addition, the same model achieved the highest accuracy among other models in two studies (89.7% and 94.87%) [29,33,34]. Spoon et al. were the first to use a handwriting dataset for the prediction of dyslexia disorder. They proposed a CNN

model to automatically recognize dyslexic children based on their handwriting [25]. They achieved an accuracy of 55.7 ± 1.4% by employing five-fold CV. Then, they developed their model further [26], reaching an accuracy of 77.6%. Yogarajah et al. [28] utilized handwriting images from children's notebooks as a dataset to build a CNN model, achieving good accuracy of 86.14 ± 1.02%. Two consecutive studies [4,29] used the same datasets from NIST 2019 to build a prediction model. The first used different CNN models, such as CNN-1, CNN-2, CNN-3, and LeNet-5, to compare the prediction performance. All models showed an accuracy of more than 87%, with CNN-1 providing the highest accuracy. The second study utilized a LeNet-5 model and achieved an outstanding accuracy of 95%, which is the highest accuracy for predicting dyslexia using a handwriting dataset. Regarding the performance of DL models for predicting dyslexia using MRI and fMRI, Zahia et al. [37] presented a 3D CNN model that achieved an accuracy of 72.7%, a sensitivity of 75%, an F1-score of 67%, a precision of 60%, and a specificity of 71.4%. Muniyandi and Usman [12] proposed a two-path cascading CNN model and achieved an accuracy of 84.6%, a specificity of 78.2%, and a sensitivity of 76.5%. In addition, Silva et al. [39] developed a CNN model and obtained the highest accuracy of 94%. Karim et al. utilized an MLP classifier to monitor accuracy with eyes opened and closed, reporting accuracies of 86% and 84.95% for eyes opened and closed, respectively.

Eye-tracking datasets have been used to build a CNN model for dyslexia prediction, which showed good results [40,41,43]. Nerušil et al. [40] achieved the highest accuracy of 96.6%, while Vags et al. [43] achieved an accuracy of 87%. Moreover, Appadurai and Bhargavi [41] achieved an accuracy of 82% for scan path images and 87% for fixation images. Table 3 summarizes the performance of the predictive model of each study in this review.

**Table 3.** Summarizes the performance of the predictive model of each study in this review.

| Reference | Datasets | DL Model | No. of Subjects | Performance |
| --- | --- | --- | --- | --- |
| [30] | Test score | ANN | Not mentioned | Accuracy: 75% |
| [32] | EGG | MLP | N = 6 kids<br>Normal = 3<br>Dyslexics = 3 | Accuracy: 86% for eye opened and 85% for eye closed. |
| [38] | fMRI and DTI | ANN | N = 56 kids<br>aged (9–12 years) | Accuracy: 94.8%, Sensitivity: 94.7%, Specificity: 95% |
| [33] | EGG scan | ANN | N = 80 kids<br>(7 to 13 ages) | Accuracy: 89.9% |
| [34] | EGG | ANN | N = 32<br>Normal = 15<br>Dyslexics = 17 | Accuracy: 78% |
| [25] | Handwriting image | CNN | N = 88<br>Normal = 62<br>Dyslexics = 11 | Accuracy: 55.7% |
| [26] | Handwriting image | CNN | N = 100 | Accuracy: 77.6% |
| [27] | Handwriting image | ANN/MLP | N = 30 | Accuracy: 73.33% |
| [12] | MRI | CNN | N = 45 | Accuracy: 73.2% |
| [37] | fMRI | 3D CNN | N = 66 children<br>(9 and 12 years) | Accuracy: 72.7%, F1 score: 67%, Sensitivity: 75.0%, Specificity: 71.4%, Precision: 60%% |
| [28] | Handwriting image | CNN | N = 54 children<br>(267 samples) | Accuracy: 86.14 ± 1.02% |
| [35] | EEG | CNN | N = 48<br>32 skilled readers<br>16 dyslexic readers | The study did not mention the accuracy but stated the effectiveness of CNN for eliciting informative features to diagnose dyslexia. |

**Table 3.** *Cont.*

| Reference | Datasets | DL Model | No. of Subjects | Performance |
|---|---|---|---|---|
| [4] | Handwriting image | CNN | Normal = 78,275 letters<br>Dyslexic = 52,196 letters | Accuracy of CNN1: 86%<br>Accuracy of CNN2: 87%<br>Accuracy of CNN3: 86.5%<br>Accuracy of LeNet-5: 86% |
| [29] | Handwriting image | CNN | | Accuracy: 95.34% |
| [40] | Eye tracking | CNN | N = 185<br>88 low risks<br>97 high risks | Accuracy: 96.6% |
| [39] | fMRI | CNN | N = 32 children<br>16 typical readers<br>16 dyslexic readers | Accuracy: 94.8% |
| [41] | Eye tracking | CNN | N = 185<br>97 dyslexics<br>88 non-dyslexics | Accuracy of CNN: 87% |
| [43] | Eye tracking | CNN (VGG 16) | N = 30<br>15 dyslexics<br>15 normal | Accuracy: 87% |
| [44] | EOG | CNN | N = 33<br>20 dyslexics<br>13 normal | Accuracy for horizontal channel EOG signals: 98.70%<br>Accuracy for vertical channel EOG signals: 80.94% |

## 5. Discussion and Challenges

Dyslexia disorder is a learning difficulty that hinders an individual's learning skills. Commonly, this disorder is pointed to as a neurological trouble that complicates remembering and addressing information in dyslexics. Artificial intelligence methods (ML and DL) have been vastly utilized for the prediction of dyslexia over recent years. The detection processes involved successive steps starting from data acquisition and preprocessing, then the extraction and selection of features, and thereafter the training process, to the model estimation process. The prediction of dyslexia disorder based on ML, particularly DL methods, was successful in the studies included in this review, achieving high accuracies of 94.8% [34], 94.8% [35], 95.34% [25], and 96.6% [36], respectively. CNN models were the most commonly used models for dyslexia prediction and showed the highest accuracy. However, there were some challenges.

The acquisition of dyslexia datasets is not easy. The studies that utilized handwriting datasets encountered difficulties in acquiring data from dyslexia children [25,26] in addition to their limited size. Furthermore, as shown in Table 3 (column three), the number of subjects did not exceed 185, but DL models require large-scale data to improve predictive model performance. In addition, the high cost of brain-imaging technology limits its usage, and, therefore, some studies tended to use cheaper EEG and eye-tracking systems.

Often, CNN models lead to overfitting problems in the training phase, where the accuracy of the model is high with the training set but low with the testing set due to involving a large number of parameters [47]. Moreover, the dyslexia datasets suffer from their small scale, leading to overfitting. To avoid this, several techniques have been suggested, such as:

- The dropout technique that commonly employs the method of generalization. Throughout each training period, the neurons are randomly eliminated. In doing so, the power of feature selection is divided uniformly over the entire group of neurons, and the model is forced to learn multiple independent features [48].
- Data augmentation: Training the model on a substantial amount of data is the simplest method for avoiding overfitting [49]. Several strategies are employed to augment

the size of the training dataset, such as cropping, translation, and rotation. Rotation and noise injection techniques have been used in the study [29]; they contributed to enlarging the training size dataset and solved the imbalanced class problem.

The dataset preprocessing step is a critical step, and the formative features extracted from datasets using DL models, especially CNNs, depend on good processing, which affects the classification accuracy. The better and more accurate the processing, the better the performance of the model.

Hyper-parameter selection has a substantial effect on CNN performance. Any variation in the values of the hyper-parameters will affect the CNN's overall performance. Consequently, proper parameter selection is a crucial factor that should be considered throughout the creation of optimization schemes [48].

Effective CNN training necessitates powerful hardware resources, such as GPUs, which are robust concerning memory usage and processing speed. Implementing DL models on a system with these hardware resources will increase the classification speed for dyslexia.

The classification of dyslexia using DL methods requires more attention and research. In particular, DL models have a strong ability to extract features from unstructured data and, thus, to perform accurate classification. Moreover, some tools may be developed to assist in diagnosing dyslexia, such as handwriting images, which were first utilized in 2019 for prediction.

## 6. Conclusions

Dyslexia is a learning difficulty that impacts the reading, writing, spelling, and dictation processes. It usually results from a deficit in the language phonological component, which is often unforeseen by teachers or parents of kids who suffer from this disorder. About 10% of the world's population suffers from this disorder, and it is important to discover it early to reduce its impact and improve the skills of dyslexics. Researchers have suggested multiple techniques for identifying dyslexia in children. Recently, DL methods have contributed significantly to diagnosing dyslexia. This review summarized the dyslexia detection techniques that have employed DL approaches. Furthermore, this review investigated the significant factors related to dyslexia prediction, seeking to help researchers build a predictive model with good accuracy. In the future, more attention could be given to the collection of dyslexia-related datasets.

**Funding:** This research received no external funding.

**Data Availability Statement:** This review paper, as we see, does not require collecting new data, it just analyzes the data related to dyslexia according to 19 papers. The handwriting dataset related to dyslexia it is available at: https://www.kaggle.com/datasets/drizasazanitaisa/dyslexia-handwriting-dataset. The other studies, which have used brain screening data (fMRI & MRI) and eye-tracking data are not available on the common database, therefore could be available on request from the authors.

**Conflicts of Interest:** The authors declare no conflict of interest.

## Abbreviations

| | |
|---|---|
| ICT | Information and Communication Technologies |
| CASL | Comprehensive Assessment of Spoken Language |
| CTOPP-2 | Comprehensive Test of Phonological Processing-2 |
| WRMT | Woodcock Reading Mastery Test |
| GSRT | Gray Silent Reading Test |
| MRI | Magnetic Resonance Imaging |
| DTI | Diffusion Tensor Imaging |
| GPUs | Graphics Processing Units |
| AI | Artificial Intelligence |



| | |
|---|---|
| ANN | Artificial Neural Networks |
| MLP | Multi-layer Perception |
| PRISMA | Preferred Reporting Items for Systematic review and Meta-Analyses |
| ERP | Event-Related Potentials |
| RFE-CV | Recursive feature Elimination with Cross-Validation |
| BET | Brain Extraction Tool |
| TBSS | Track-Based Spatial Statistics |
| OCR | Optical Character Recognition |
| ADM | Association of Dyslexia Malaysia |
| SSA | Singular Spectrum Analysis SSA |
| PCA | Principal Component Analysis |
| ML | Machine Learning |
| SVM | Support Vector Machine |
| DL | Deep Learning |
| 3D | Three-Dimensional. |
| TEWL | Test of Early Written Language |
| fMRI | Functional MRI |
| EEG | Electroencephalography |
| IEEE | Institute of Electrical and Electronic |
| CNN | Convolutional Neural Network |
| DNN | Deep Neural Network |
| RNNs | Recurrent Neural Networks |
| PCA | Principal Component Analysis |
| FA | Fractional Anisotropy |
| FEAT | FMRI Expert Analysis Tool |
| FDT | FMRIB Diffusion Toolbox |
| EOG | Electrooculogram |
| STFT | Short-Time Fourier Transform |
| PSD | Power Spectral Density |

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
