# Peer review of "Deep Learning Applications for Dyslexia Prediction"

_applsci, doi:10.3390/app13052804_

Round 1

Reviewer 1 Report

Overall the authors wrote a good article summarizing DL findings in dyslexia detection. It is generally well-written, but there is some confusion in some terms. I am sure authors can improve the text to make it easier to follow and understand for both DL experts and clinicians. I hope my comments will help to improve the article, since the authors already performed a good job and there are minor things to correct.

On page 3, the authors state that: "ANNs are multi-layer fully connected neural nets, which consist of three strata: input strata, several hidden strata, and output strata. In each layer, every node is linked to every other node in the next layer". This assumption is not correct since RNN or CNN are also part of the DL frame and can be designed as not fully connected.

At the same page authors state: ". The essential processes in DL are feature extraction and selection". The reference you cite next to the text do not express this. Are authors sure that the essential processes are "feature extraction" and "selection"? Those two processes are performed inside the algorithm, but apart from different algorithms that use XAI (explainable AI) the majority of the DL algorithms perform the feature selection/extraction internally. Maybe you want to express that the algorithm performs those two functions and that is crucial for the algorithm to work well, then the text is quite confusing for me.

Reading the rest of the article, I think authors may want to express that a feature extraction or selection step before applying the method is usually recommended. If this is what authors will to express, I agree, but depending on the dimensionality of the data, DL should be able to perform by itself the extraction or selection of features.

On page 9, "Feature extraction refers to the process of converting raw data into numerical (or categorical) features that could be handled in addition to keeping the information in the basic dataset." I think the sentence is quite confusing.

Same page: "The feature extraction and selection operation is a significant task in dyslexia classification and prediction because the number of features is largely related to computational complexity". I'm not sure if significant is the correct word. I just mention it if authors feels that it could be better described.

I checked a few methods from table 2 and all were wrongly described. Most of the articles don't use DL to perform any feature selection/extraction, instead they apply a prior algorithm (non DL in most cases) . For eample in article 33, they use SPM software to extract activations. Keras and Tensorflow are used for prediction. I would encourage authors to understand what feature selection/extraction is, and to rewrite this table. I know there is general confusion, and so, I encourage the authors to write a short paragraph or even a short sentence saying that some preprocessing is needed in these datasets, previous to use the DL algorithm if they find it useful.

In page 11, authors say that "Although some studies used ANNs to deal with EGG datasets and achieved high accuracy (96%) [29], others achieved low accuracy (78%) [30].". A 78% of accuracy I would not say it is low, specially considering the difficulty of the task. I would change "low" by "lower" or "modest". 

On the same page, authors mention the concept of training and testing data and later say, "the remainder  was allocated for validation and testing". I would include a small description of what validation data is. I say this because in DL validation data is often the data used to evaluate the performance of the trained model during training, and test data is the evaluation of the final model. But in ML or other fields, validation is the evaluation of the final model. A simple explanation of those terms would be good for the reader.

In addition, it would be good to add a small explanation (or a column in table 3) telling whether in imbalanced datasets the Accuracy corresponds to Balanced Accuracy, or any correction. Also, table 3 has specific information for some studies (study 39 specifies VGG6 algorithm, which in fact does not exist, it is VGG16; in other studies the term ANN, CNN, and MLP are mentioned. I don't think this adds much information.

Page 13: "the datasets suffer from their scale". Does authors refer to dimensionality or small sample size? I think it is correct, but just for you to assure.

Author Response

Dear respected Reviewer:
Thank You for your helpful suggestions that have helped us to improve the manuscript.

Comments: On page 3, the authors state that: "ANNs are multi-layer fully connected neural nets, which consist of three strata: input strata, several hidden strata, and output strata. In each layer, every node is linked to every other node in the next layer". This assumption is not correct since RNN or CNN are also part of the DL frame and can be designed as not fully connected

Reply: Thank you for pointing this out. we corrected that

Comments: At the same page authors state: ". The essential processes in DL are feature extraction and selection". The reference you cite next to the text do not express this. Are authors sure that the essential processes are "feature extraction" and "selection"?

Reply: The reference [13] stated that "The essential processes in DL are feature extraction and selection" in page 34

Comments: I checked a few methods from table 2 and all were wrongly described. Most of the articles don't use DL to perform any feature selection/extraction, instead they apply a prior algorithm (non DL in most cases) . For eample in article 33, they use SPM software to extract activations. Keras and Tensorflow are used for prediction.

Reply: Thank you for illustrating this. we corrected that.

Comments: In page 11, authors say that "Although some studies used ANNs to deal with EGG datasets and achieved high accuracy (96%) [29], others achieved low accuracy (78%) [30].". A 78% of accuracy I would not say it is low, specially considering the difficulty of the task. I would change "low" by "lower" or "modest".

Reply: we change that to lower.

Comments: On the same page, authors mention the training and testing data concept and later say, "the remainder was allocated for validation and testing". I would include a small description of what validation data is.

Reply: Thank you for pointing this out. We introduce a brief definition of validation. 

Comments :Page 13: "the datasets suffer from their scale". Does authors refer to dimensionality or small sample size?

Reply: we mean size, and We have rewritten it as "suffer from their small scale"

Comments: study 39 specifies VGG6 algorithm, which in fact does not exist, it is VGG16;

Reply: we corrected that,

Reviewer 2 Report

The number of papers utilized in the paper is not enough for the review/survey paper. Please be more comprehensive. The language of the manuscript should be improved.

Author Response

Thank You for your helpful suggestions that have helped us to improve the manuscript.

Comments: The number of papers utilized in the paper is not enough for the review/survey paper. Please be more comprehensive. The language of the manuscript should be improved.

Reply: Thank you for revising. Some sections have been expanded in this Manuscript and according to that, 8 references have been added. Also, this Manuscript has been revised and subjected to proofreading before being submitted.

Reviewer 3 Report

This manuscript presents a review of research evaluating deep learning in the prediction of dyslexia. While the extraction of papers followed PRISMA protocols and the methods were well-described, there are some weaknesses that need to be addressed.

The Introduction presents a very basic grasp of dyslexia and related neuroimaging research, with some inaccuracies related to methods of diagnosis. Specifically:

· Page 1, Final Paragraph. The sentence discussing the availability of screening measures online is problematic. Many of the mentioned screening tests may be available via online formats, but still have regulations on who is an authorized user and who can purchase it. Further, it’s stated that the tests are available on the web “due to ICT that authorized the digitalization of the tests”. ICT is never defined and the citation provided [5] is to a review paper. It would be better practice to directly cite the authorization. Finally, medical devices are used in research to help better understand the processes underlying dyslexia, but they are not used in diagnosis. The criteria for a practitioner to diagnose dyslexia are specific to performance on specific tests of cognitive and academic functioning in connection with ruling out other potential causes of poor reading achievement.

·         Figure 1 is over-simplistic and does not reflect the highly complex interactions within and across hemispheres related to reading. Recommend it be cut.

·         Page 2. Three different types of dyslexia are mentioned at the end of the top paragraph. The citation [6] is not correct. Additionally, subcategories of dyslexia are not consistently described within the literature (i.e., other researchers suggest a dual-route model) – we caution against focusing on subcategories for the purpose of this manuscript.

·         Page 3, Final Paragraph before “2. Related Work”. Suggest rephrasing the first two sentences: “This paper is organized into 6 parts. Following this initial introduction, the second section provides…”

In the Discussion and Challenges section, it was hoped that there would be more information on which prediction variables lead to higher accuracies within the models. Further, the difficulties in obtaining imaging data on patients as well as the reduction in predictive value outside of the training set are briefly mentioned. As these factors are highly relevant to practitioners, these should be elaborated on.

We recommend that the authors bring in a clinical expert on the diagnosis of dyslexia to ensure that the clinical aspects of dyslexia diagnosis and treatment are accurately represented in the Introduction and Discussion/Conclusion.

Author Response

Dear respected Reviewer:
Thank You for your helpful suggestions that have helped us to improve the manuscript.

Comments: Page 1. Final Paragraph. The sentence discussing the availability of screening measures online is problematic. Many of the mentioned screening tests may be available via online formats, but still have regulations on who is an authorized user and who can purchase it. Further, it’s stated that the tests are available on the web “due to ICT that authorized the digitalization of the tests”. ICT is never defined and the citation provided [5] is to a review paper. It would be better practice to directly cite the authorization.

Reply: Thank you for pointing this. Yes, the dyslexia tests are different among them and subject to some criteria such as age, Administration Time, and price. But in our paper, we are interested to illustrate to the readers there are available methods for the prediction of dyslexia, and we mentioned some of them before moving to the advanced prediction by deep learning. We added an abbreviations section above the introduction and mentioned ICT, defining it as displayed in the paper. In addition, we changed the citation to point to the original paper, not the review.

Comments:      Figure 1 is over-simplistic and does not reflect the highly complex interactions within and across hemispheres related to reading. Recommend it be cut

Reply: We re-drew the figure to become more formal, and we preferred to add this figure to our paper to let the readers know the different skills controlled by the left and right hemispheres.

Comments: Three different types of dyslexia are mentioned at the end of the top paragraph. The citation [6] is not correct. Additionally, subcategories of dyslexia are not consistently described within the literature (i.e., other researchers suggest a dual-route model) – we caution against focusing on subcategories for the purpose of this manuscript.

Reply: Thank you for pointing this, the citation has been corrected now

Comments: Page 3, Final Paragraph before “2. Related Work”. Suggest rephrasing the first two sentences: “This paper is organized into 6 parts. Following this initial introduction, the second section provides…”

Reply: Thank you for your suggestion, we paraphrased it now

Comments: In the Discussion and Challenges section, it was hoped that there would be more information on which prediction variables lead to higher accuracies within the models

Reply: we expanded this section to provide more information for researchers and introduced more challenges related to DL models

Round 2

Reviewer 2 Report

All the earlier concerns have been properly addressed.

Author Response

Thank you for helping us to improve our paper.

Reviewer 3 Report

The only remaining significant concern is that when the list of screening tests are mentioned and it is stated that they have been made available on the web, it makes it sound like anyone can use them. We recommend adding something to clarify that most of those tests can only be purchased by qualified users (those with advanced training in psychological assessment).

Author Response

Dear respected Reviewer:
Thank You for your helpful suggestions that have helped us to improve the manuscript.

comments:
The only remaining significant concern is that when the list of screening tests are mentioned and it is stated that they have been made available on the web, it makes it sound like anyone can use them. We recommend adding something to clarify that most of those tests can only be purchased by qualified users (those with advanced training in psychological assessment).

Reply:
Thank you, text to clarify this has been added.